# Effectiveness and promising behavior change techniques of interventions targeting energy balance related behaviors in children from lower socioeconomic environments: A systematic review

**Manou Anselma**[ID]*, **Mai J. M. Chinapaw**[ID], **Daniëlle A. Kornet-van der Aa, Teatske M. Altenburg**

Department of Public and Occupational Health, Amsterdam Public Health Research Institute, Amsterdam UMC, Vrije Universiteit Amsterdam, Amsterdam, The Netherlands

* m.anselma@amsterdamumc.nl

## Abstract

This systematic review aims to summarize the evidence regarding the effectiveness of interventions targeting energy balance-related behaviors in children from lower socioeconomic environments and the applied behavior change techniques. The literature search was conducted in Cochrane, Embase, Psycinfo and Pubmed. Articles had to be published between January 2000 and September 2019. Studies were included that i) targeted dietary behavior, physical activity and/or sedentary behavior; ii) had a controlled trial design; iii) included children aged 9–12 years old; iv) focused on lower socioeconomic environments; and v) took place in upper-middle or high income countries. Two independent researchers extracted data, identified behavior change techniques using the Behavior Change Technique Taxonomy v1, and performed a methodological quality assessment using the quality assessment tool of the Effective Public Health Practice Project. We included 24 studies, of which one received a high and three a moderate quality rating. Demonstration, practice and providing instructions on how to perform a behavior were the most commonly applied behavior change techniques. Seven studies reported significant beneficial intervention effects: five on physical activity, one on physical activity and sedentary behavior and one on dietary behavior. When comparing effective versus non-effective interventions, and comparing our review to previous reviews focusing on children from the general population, similar behavior change techniques were applied. More high quality research is needed to evaluate the effectiveness of interventions and their behavior change techniques targeting children of low socioeconomic environments.

PROSPERO registration number: CRD42016052599

**Data Availability Statement:** All relevant data are within the manuscript and its Supporting Information files.

**Funding:** This study is part of the Kids in Action study, which was funded by FNO (grant number 101569; https://www.fnozorgvoorkansen.nl/). The funder had no role in the study design, data collection and analysis, decision to publish, or preparation of the manuscript.

**Competing interests:** The authors have declared that no competing interests exist.

## Introduction

Obesity in children remains a major public health problem, with overall rates still rising [1], especially in children from families with a low socioeconomic position [2, 3]. Children with overweight or obesity are more likely to maintain overweight or obese into adulthood [4]. Treating obesity has shown to be extremely difficult, amplifying the need for early prevention [5]. When children adopt healthy energy balance-related behaviors at a young age, they are more likely to continue these habits into adolescence and adulthood [6]. Therefore, adopting healthy habits at an early age is an important public health target [7, 8]. This is especially true for children living in low socioeconomic neighborhoods, where many children experience multiple barriers to engage in healthy behavior such as lack of finances and transport, and are therefore at an increased risk of developing obesity [9, 10]. Hence, effective interventions are needed that stimulate healthy energy balance-related behaviors in children from low socioeconomic environments to reduce health inequalities between children from lower and higher socioeconomic positions. Previous systematic reviews focused on children from all socioeconomic positions [11], on adolescents [12], on children from a specific ethnicity [13, 14] or were limited to specific intervention designs such as family-based [15], school-based [16], or policy interventions [17]. Effective components of obesity prevention interventions in children identified in systematic reviews include school policies regarding the availability of foods and beverages meeting nutritional standards; targeting multiple behaviors and system levels; encouragement of environments and cultural practices at school and home that support healthy behavior; education of children, parents and teachers on healthy nutrition and physical activity; improvement of physical education programs and physical activity possibilities in policy and practice [11, 16–20]. Previous studies have also shown that energy-balance related behaviors and its determinants may manifest themselves differently in children from different socioeconomic levels [21–25]. To reduce health inequalities between children from lower and higher socioeconomic environments, more insight is therefore needed in interventions and intervention strategies that are specifically effective in stimulating healthy energy balance-related behaviors among children from low socioeconomic environments.

The current review aims to summarize the effectiveness of interventions targeting physical activity, sedentary behavior and/or dietary behavior among 9–12 year old children from low socioeconomic environments. An important note is that these interventions target children attending schools or living in neighborhoods defined as 'disadvantaged' or 'low-income', indicating that a substantial percentage of children in these schools or neighborhoods have a low socioeconomic position. The age group of 9-12-year olds was chosen because the transition phase from mid-childhood into adolescence is a critical period, due to biological changes as well as changes in the social and physical environment due to a change in school environment [26, 27]. A second aim was to identify effective behavior change techniques using the Behavior Change Technique (BCT) Taxonomy v1 [28]. Knowledge of BCTs used in interventions that are effective in improving health behaviors in children from low socioeconomic environments is important to inform future intervention development and improve the health of the children who mostly need it.

## Materials and methods

The protocol for this review was registered in PROSPERO (registration number: CRD42016052599). The Preferred Reporting Items for Systematic reviews and Meta-Analyses (PRISMA) statement was used to plan, conduct and transparently report this systematic review [29].

## Literature search

A systematic literature search was conducted in four databases: Cochrane, Embase, Psycinfo and Pubmed. Articles between January 2000 and September 2019 were included. The search was limited to articles published after 2000, to include interventions that are relevant for today's society. The search terms were related to health behaviors (physical activity, sedentary behavior and/or dietary behavior), health promotion, study design (controlled trial, evaluation, community or school based), socioeconomic position, and children. The full search strategy can be found in S1 Table. Studies were included that i) targeted physical activity, sedentary behavior or dietary behavior as an outcome; ii) had a controlled trial design; iii) included children aged 9–12 years old (average age of total sample or a subgroup analysis); iv) took place in low socioeconomic environments; and v) took place in upper-middle- or high-income countries. Socioeconomic environments were indicated by terms related to low-income, deprived, disadvantaged, low socioeconomic status or position. Upper-middle- (gross national income per capita between $3,896 and $12,055) or high-income countries (gross national income per capita of $12,056 or more) were defined according to criteria of the World Bank [30]. Additionally, studies had to be written in English and published in a peer reviewed scientific journal. Studies that focused on specific populations such as children with obesity, clinical samples or studies that took place in remote areas, were excluded.

## Selection process and data extraction

Two independent researchers screened all titles and abstracts retrieved from the databases (MA and DA or MA and TA). When discrepancies occurred, a third reviewer (TA or DA) was consulted, and when discrepancies could not be solved a fourth researcher (MC) was consulted. Full texts were screened by MA, and TA or DA. In case of discrepancies or uncertainties, a third and/or fourth reviewer was consulted. In case information was missing and there was a reference to a protocol paper, the protocol paper was used to retrieve the required information.

MA and TA independently extracted data, using a standard data extraction form. Inconsistencies were discussed afterwards until consensus was reached and if needed MC was consulted. Information on participant characteristics (sample size, gender, ethnicity, mean age), intervention strategies, intervention setting, intervention duration and follow-up (number of weeks after completion of intervention), control group, outcome measures and results (if reported β and 95% confidence intervals) were extracted. Results were reported separately for physical activity, sedentary behavior or dietary behavior. If analyses were stratified for gender, this was included as well to gain more insight in gender-specific intervention effects. An intervention was scored as effective when a beneficial intervention effect was obtained on at least 75% of the outcomes within that behavior (similar to e.g. Van Ekris et al., 2016 and Haynes et al., 2018 [31, 32]). For example, if a study measured eight different outcomes related to physical activity, six had to show a beneficial intervention effect for the study to be considered effective in improving physical activity.

MA and DA independently identified BCTs applied in all studies using the BCT Taxonomy v1 [28]. If needed, TA was consulted to resolve discrepancies. The BCT Taxonomy v1 consist of 93 BCTs clustered in sixteen groups. These BCTs can be used to classify components of behavior change interventions.

## Quality assessment

MA and MC conducted a methodological quality assessment using the quality assessment tool of the Effective Public Health Practice Project [33]. Studies were rated on eight items: *Selection*

*bias*, *Study design*, *Confounders*, *Blinding*, *Data collection methods*, *Withdrawals and drop-outs*, *Intervention integrity* and *Analyses* (see S2 Table for all items and sub-items). When needed, references to protocol papers or validity and reliability studies were checked. As all included studies were controlled trials, the item *Study design* was always strong and was only included to separate controlled trials from randomized controlled trials. The items *Selection bias*, *Confounders*, *Blinding*, *Data collection methods* and *Withdrawals and drop-outs*, all consisted of two sub-items and were labelled as strong if both sub-items were rated as strong. If both sub-items were weak, a weak score was given; if only one item was strong, a moderate score was given. The item *Analyses* consisted of four sub-items. If all four sub-items were strong, the item was labelled as strong; if three sub-items were strong, a moderate score was given; if two or less sub-items were strong, a weak score was given. Both assessors independently rated the included studies and afterwards inconsistencies were discussed until consensus was reached.

## Results

Fig 1 presents the flowchart of included studies. The search resulted in 25,146 items matching our search criteria (2,623 from Cochrane, 3,673 from Embase, 3,462 from Psycinfo, 15,388 from Pubmed). After removing duplicates, titles and abstracts of 17,302 items were screened and subsequently 74 full-text studies were assessed. After screening full texts and strictly checking the inclusion criteria by the third and fourth assessor (11 studies), 26 studies evaluating 25 interventions were included in the review. Reasons for exclusion of studies in this last phase were: only assessing attitudes and not behaviors, mean participant age not being between 9–12 years, focusing on specific groups such as children with obesity, or not including a control group.

### Study characteristics and quality assessment

Table 1 presents the study and intervention characteristics. The sample size of the 26 included studies varied from 51 [34] to 3,463 [35] at baseline, with eight studies having hundred or less participants in the intervention group [34, 36–42]. The intervention duration varied from four weeks to two years. Two studies targeted girls only [34, 42], the other 24 studies targeted both genders.

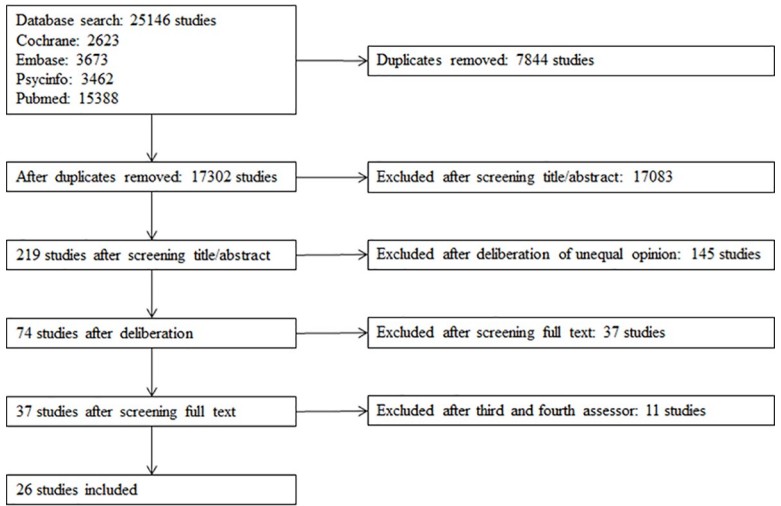

**Fig 1. Flowchart of included studies.**

**Table 1. Study and intervention characteristics of the included studies.**

| Study; design; intervention | Participants | Country, setting, indicator for lower socioeconomic environment | Duration intervention | Follow-up | Description control group |
|---|---|---|---|---|---|
| Alaimo et al., 2015 [43]; CT[a]; Project FIT | **Year 1:** <br> **IG**: N = 302; mean age 9.6±0.9(60); 46% Hispanic, 43% African American <br> **CG**: N = 108; 62% Hispanic, 18.5% African American <br> **Year 2:** <br> **IG**: N = 302; mean age 9.7±0.9(60); 47% Hispanic, 42% African American <br> **CG**: N = 103; 70% Hispanic, 9% African American | USA; school and community; lowest-income neighborhoods in the school district | Continuous in school year—study duration 2 years | NA | No intervention |
| Ashfield-Watt et al., 2008 [51]; RCT; no name | **IG**: N = 1010; mean age 8.7±1.1; 50.3% female; 2.1% Asian, 5.5% European, 23.3% New Zealand Maori, 57.8% Pacific People, 11.3% Mixed/Other <br> **CG**: N = 890; mean age 8.8±1.2; 49.9% female; 1.2% Asian, 3.5% European, 22.8% New Zealand Maori, 57.3% Pacific People, 15.1% Mixed/Other | New Zealand; school; suburbs with a high level of economic deprivation (most schools were from an area where 45% of children are classified as the most deprived in the country) | 10 weeks | 6 weeks | No intervention |
| Bastian et al., 2015 [52]; CT[a]; APPLE schools | Mean age 10.9 years <br> 2009: <br> **IG:** N = 198; 47.2% female <br> **CG:** N = 454; 50.8% female <br> 2011: <br> **IG:** N = 196; 51.0% female <br> **CG:** N = 309; 49.1% female | Canada; school; socioeconomically disadvantaged neighborhoods | Continuous in school year—study duration 2 years | NA | No health facilitator; no APPLE promotion materials; only materials to implement Alberta health's provincial Healthy Weights Initiative. |
| Beyler et al., 2014 [44]; RCT; Playworks | **IG**: N = 1285; 52.4% female; 31.5% Black/African American, 33.0% Hispanic/Latino, 27.2% White, 23.7% Asian/Native Hawaiian, 9.0% American Indian/Alaskan Native <br> **CG**: N = 993; 50.6% female; 30.5% Black/African American, 47.4% Hispanic/Latino, 21.9% White, 12.9% Asian/Native Hawaiian, 6.4% American Indian/Alaskan Native | USA; school; low-income schools in urban areas, in which at least 50% of students qualify for FRP | 1 school year (±7 months) | NA | No intervention |
| Bohnert and Ward, 2013 [42]; RCT; Girls in the Game (GIG) | **IG**[b]: N = 52; mean age 9.0±0.9 years; 100% female; 35.3% African-American, 60.8% Latina, 3.9% Caucasian <br> **CG**[b]: N = 24; mean age 9.4±1.1 years; 100% female; 37.5% African-American, 58.3% Latina, 4.2% Caucasian | USA; school (after school hours); underserved urban low-income communities, with low-income status ranging from 72–98% as indicated by city report | 30 weeks | NA | No intervention; only participating in the health festivals. |
| Breslin et al., 2012 [54]; CT; Sport for LIFE | **IG and CG**[b]: mean age 9.1±0.4 years; 51.7% female <br> **IG**[b]: N = 209 <br> **CG**[b]: N = 207 | Northern Ireland; school; schools scoring worse than average on: the proportion of free school meals (>15%), the proportion of pupils taking the transfer test to secondary/grammar school level (<65%) and attaining a grade A (<25%) | 12 weeks | NA | No intervention |
| Colín-Ramírez et al., 2010 [55]; RCT; RESCATE | **IG and CG**[b]: mean age 9.4±0.7 years <br> **IG**[b]: N = 245; 44% female <br> **CG**[b]: N = 253; 53% female | Mexico; school; schools of low socioeconomic status in Mexico City | 12 months | NA | No intervention |
| Dunton et al., 2015 [41]; RCT; no name | **IG**: N = 54; mean age 10.3±1.4 years; 52.7% female; parents 3.8% African-American/Black, 1.9% Asian, 73.1% Hispanic/Latino, 15.4% White, 5.8% Other <br> **CG**: N = 57; mean age 10.1±1.2 years; 58.6% female; parents 7.5% African-American/Black, 1.9% Asian, 52.8% Hispanic/Latino, 32.1% White, 5.7% Other | USA; school (after school hours); schools with approximately 50% of enrolled students participating in the FRP program | 4 months | 6 weeks | No intervention |

*(Continued)*

**Table 1.** (Continued)

| Study; design; intervention | Participants | Country, setting, indicator for lower socioeconomic environment | Duration intervention | Follow-up | Description control group |
|---|---|---|---|---|---|
| Gatto et al., 2017 [49]; RCT; LA Sprouts | **IG**[b]: N = 172; mean age 9.3±0.9; 47.7% male; 89.0% Hispanic/Latino | USA; school (after school hours); 75% of students participating in the FRP program | 12 weeks | NA | No intervention |
| | **CG**[b]: N = 147; mean age 9.3±0.9; 48.3% male; 88.8% Hispanic/Latino | | | | |
| Gittelsohn et al., 2010 [40]; CT; Healthy Foods Hawaii (HFH) | **IG and CG**: 64% Native Hawaiian or other Pacific Islander | USA; community; >75% of the population is below the poverty level | 9–11 months | NA | No intervention |
| | **IG**: N = 64 child-caregiver dyads; mean age children 9.8±1.3 years; 50% female | | | | |
| | **CG**: N = 53 child-caregiver dyads; mean age children 9.9±1.4 years; 47.2% female | | | | |
| Harrison et al., 2006 [56]; CT; Switch Off-Get Active | **IG**: N = 182; mean age 10.2±1.2 years; 44% female | Ireland; school; areas of greatest social disadvantage according to classifications of the local health authority | 16 weeks | NA | No intervention |
| | **CG**: N = 130; mean age 10.3±0.8 years; 42% female | | | | |
| Keihner et al., 2017 [35]; RCT; Power Play! | **IG**[b]: N = 1571 | USA; school; low-resource public schools (50% of students receiving FRP) | 10 weeks | NA | Not reported |
| | **CG**[b]: N = 1892 | | | | |
| | **IG and CG**: mean age 9.7±0.7 years; 41.4% Hispanic, 25.3% other/mixed race, 12.5% non-Hispanic White, 9.1% Asian, 8.9% non-Hispanic Black | | | | |
| Lent et al., 2014 [45]; RCT; Snackin' Fresh intervention | **IG**: N = 435; mean age 10.97±1.02 years; 55.4% female; 46.2% Black/African American, 0.5% White, 43.2% Hispanic/Latino, 0.5% Asian, 0.2% Native American/Alaskan native, 9.4% other/mixed/unknown | USA; schools and corner stores (i.e. school-store cluster); schools were located in low-income neighborhoods and had >50% of students qualifying for FRP | 2 years | NA | No intervention |
| | **CG**: N = 332; mean age 10.99±0.92 years; 57.8% female; 38.3% Black/African American, 13.2% White, 16.2% Hispanic/Latino, 15.9% Asian, 1.5% Native American/Alaskan native, 15% other/mixed/unknown | | | | |
| Madsen et al. 2013 [39]; RCT; America SCORES | **IG**: N = 82; mean age 9.8±0.6 years; 38% female; 14% African-American, 36% Asian, 38% Latino, 0% White, 13% other | USA; school (after school hours); 61% of students were eligible for FRP (range 44%-89%) | 2x 12 weeks | NA | No intervention |
| | **CG**: N = 74; mean age 9.8±0.7 years; 42% female; 11% African-American, 27% Asian, 45% Latino, 1% White, 16% Other | | | | |
| Mendoza et al., 2017 [36]; RCT; Bicycle trains | **IG**: N = 24; mean age 9.8±0.8 years; 54.1% female; 4.2% non-Latino White, 37.5% non-Latino Black, 20.8% Latino, 12.5% Asian, 16.7% Multi-racial/Other, 8.3% missing | USA; school (before and after school hours) and community; schools of which >60% of students qualified for the FRP | ~4 weeks | NA | No intervention |
| | **CG**: N = 30; mean age 10.0±0.7 years; 73.3% female; 6.7% non-Latino White, 13.3% non-Latino Black, 33.3% Latino, 26.7% Asian, 13.3% Multi-racial/Other, 6.7% missing | | | | |
| Neumark-Sztainer et al., 2009 [46]; RCT; Ready. Set. ACTION! | **IG and CG**: N = 108; mean age 10.3±1.1 years | USA; school (after school hours); ±90% of the students qualified for FRP | 1 school year (from fall to spring) | NA | Theatre based intervention focused on environmental health issues |
| | **IG**: N = 56; 54% African-American, 13% Asian/Hmong, 7% White, 3% Hispanic, 23% Other/mixed | | | | |
| | **CG**: N = 52; 55% African-American, 15% Asian/Hmong, 7% White, 1% Hispanic, 22% Other/mixed | | | | |

*(Continued)*

**Table 1.** (Continued)

| Study; design; intervention | Participants | Country, setting, indicator for lower socioeconomic environment | Duration intervention | Follow-up | Description control group |
|---|---|---|---|---|---|
| Nollen et al., 2014 [34]; RCT; no name | **IG**: N = 26; mean age 11.3±1.5 years; 100% female; 80.8% African-American, 11.5% bi- or multi-racial, 7.7% American Indian/Alaska Native, 0.0% Asian/Pacific Islander, 7.7% Hispanic/Latina | USA; mobile phone; economically disadvantaged neighborhoods | 12 weeks | NA | Written manual, no action cues or reward setting |
| | **CG**: N = 25; mean age 11.3±1.7 years; 100% female; 86.9% African-American, 4.4% bi- or multi-racial, 4.4% American Indian/Alaska Native, 4.4% Asian/Pacific Islander, 8.0% Hispanic/Latina | | | | |
| Salmon et al. 2008 [57]; RCT; Switch-Play | Mean age boys 10.7±0.4 years, mean age girls 10.7±0.3 years | Australia; school; low socioeconomic status areas (based on socioeconomic index for areas scores) | 1 school year | 6 and 12 months | No intervention |
| | **IG** (3 groups): N = 66 (BM), N = 74 (FMS), N = 93; (BM/FMS); 50.7% female (BM), 52.6% female (FMS), 51.1% female (BM/FMS) | | | | |
| | **CG**: N = 62; 50.8% female | | | | |
| Salmon et al., 2011 [58]; RCT; Switch-2-Activity | **IG and CG:** mean age 10.3±0.6 years; 58% female | Australia; school; low socioeconomic urban areas | 7 weeks | NA | Waitlist control group |
| | **IG**: N = 467 | | | | |
| | **CG**: N = 490 | | | | |
| Slusser et al., 2013 [38]; CT; Catch Kids Club | **IG**[b]: N = 73; 58.9% female; 15.1% Hispanic/Latino, 67.1% Asian/Pacific Islander, 17.8% Other | USA; school (after school hours); schools in a district where more than two-thirds (67.8%) of the students qualified for FRP | 1 school year | NA | Other after-school programs without support for nutrition education and physical activity |
| | **CG**[b]: N = 48; 56.3% female; 39.6% Hispanic/Latino, 50.0% Asian/Pacific Islander, 10.4% Other | | | | |
| Springer et al., 2012 [47]; CT; Marathon Kids | **IG**: N = 383; mean age 9.9±0.9 years; 49.6% female; 78.6% Hispanic, 6.5% African-American, 11.7% White, 3.1% other | USA; school and community; schools with ≥60% students who are economically disadvantaged, based on criteria for classifying schools as low-income provided by the funding agency | 6 months | 2 months | No intervention |
| | **CG**: N = 128; mean age 10.0±0.8 years; 56.3% female; 76.6% Hispanic, 14.8% African-American, 3.9% White, 4.7% other | | | | |
| Trude et al., 2018 [50]; RCT; B'more Healthy Communities for Kids (BHCK) | **IG:** 9–15 year olds: N = 273 (70.7% 9–12 year olds); mean age 11.7±1.3; 54.1% female; 95.5% African-American | USA; community food environment; low-income neighborhood (>20% of residents living below the poverty line) | 2 waves: 3 phases of 2 months | NA | Waitlist control group |
| | **CG:** 9–15 year olds: N = 236 (61.8% 9–12 years old); mean age 11.9±1.6; 57.2% female; 97.5% African-American | | | | |
| Van de Gaar et al., 2014 [59]; RCT; Water campaign | Children with at least one report (parent, child, observation): | the Netherlands; school; socially more deprived neighborhoods | 1 school year | NA | Regular health promotion program |
| | **IG**: N = 504 observation report, N = 158 parent report, N = 182 child report. Based on child-report: 50.6% female, 29.7% Dutch, 13.7% Surinamese/Antillean, 32.4% Moroccan/Turkish, 24.2% other/missing. | | | | |
| | **CG**: N = 455 observation report, N = 198 parent report, N = 205 child report | | | | |
| | Based on child-report: | | | | |
| | 55% female, 17.6% Dutch, 29.3% Surinamese/Antillean, 33.2% Moroccan/Turkish, 20.0% other/missing. | | | | |

(*Continued*)

**Table 1.** (Continued)

| Study; design; intervention | Participants | Country, setting, indicator for lower socioeconomic environment | Duration intervention | Follow-up | Description control group |
|---|---|---|---|---|---|
| Vander Ploeg et al., 2014 [53]; CT[a]; APPLE schools | Mean age 10.9 years; 49.5% female | Canada; school; socioeconomically disadvantaged neighborhoods | 2 years | NA | No access to a health facilitator or health promotion materials; but they received materials to implement Alberta Health's provincial Healthy Weights Initiative (public information and education campaign) |
| | 2009 | | | | |
| | **IG:** N = 358; 47.2% female | | | | |
| | **CG:** N = 454; 50.8% female | | | | |
| | 2011: | | | | |
| | **IG:** N = 196; 51.0% female | | | | |
| | **CG:** N = 309; 49.1% female | | | | |
| Wang et al., 2019 [37]; RCT; H₂GO! | **IG[b]:** N = 51; mean age 10.0±1.1 years; 56.9% female; 11.4% White, 38.6% Black, 43.2% Hispanic/Latino, 2.3% Asian, 4.6% Multiracial/Other | USA; community-based; predominately low socioeconomic backgrounds | 6 weeks | 2 and 6 months | Standard 'Boys and Girls Clubs of America' programming |
| | **CG[b]:** N = 49; mean age 10.2±1.0 years; 34.7% female; 9.3% White, 20.9% Black, 32.6% Hispanic/Latino, 27.9% Asian, 9.3% Multiracial/Other | | | | |
| Wells et al., 2014 [48]; RCT; Healthy Gardens, Healthy Youth | **IG:** N = 115; mean age 9.5±0.7 years; 56.5% female; 67.0% White, 21.7% African-American, 8.7% Hispanic, 2.6% Asian | USA; school; ≥50% of students qualifying for FRP | 1 year | NA | Waitlist control group |
| | **CG:** N = 112; mean age 9.0±0.5 years; 56.3% female; 35.7% White, 38.4% African-American, 8.9% Hispanic, 17.0% Asian | | | | |

[a] indicates cross-sectional analysis

[b] indicates analyzed at follow-up.

BM = behavioral modification, CG = control group, CT = controlled trial, FMS = fundamental movement skills, FRP = free or reduced-price meal, IG = intervention group, NA = not applicable, RCT = randomized controlled trial.

Seventeen studies were conducted in the US [34–50], one study was conducted in New Zealand [51], two in Canada [52, 53], one in Northern Ireland [54], one in Mexico [55], one in Ireland [56], two in Australia [57, 58] and one in the Netherlands [59]. Fourteen studies clearly defined lower socioeconomic environments, such as a certain percentage of children eligible for free or reduced-price meals at school. Other studies used more general criteria such as living in 'low socioeconomic areas' or 'disadvantaged neighborhoods'. Fourteen studies were performed in the school setting [35, 42, 44, 48, 49, 51–59] and four after school time [38, 39, 41, 46]. The other studies were conducted both at school and in the community [36, 43, 47], only in the community [37, 40, 50], at school and corner stores [45], and there was one mobile application intervention [34]. Studies performed in the USA [34–50], New Zealand [51] and the Netherlands [59] described the ethnicity of their study population. The studies from Canada [52, 53], Northern Ireland [54], Mexico [55], Ireland [56] and Australia [57, 58] did not specify their study sample's ethnicity. In seventeen studies, the control group received no intervention; in three studies, the control groups received the intervention after completion of the study (i.e. waitlist control group) [48, 50, 58]; in the other studies, the control groups received part of the intervention [34, 42, 53], a different health program [52] or a program not related to health behavior [38, 46].

S3 Table presents the methodological quality rating of the included studies for each of the items of the quality assessment tool of the Effective Public Health Practice Project. Fig 2 presents an overview of the methodological quality rating per item of the included studies. Fig 3

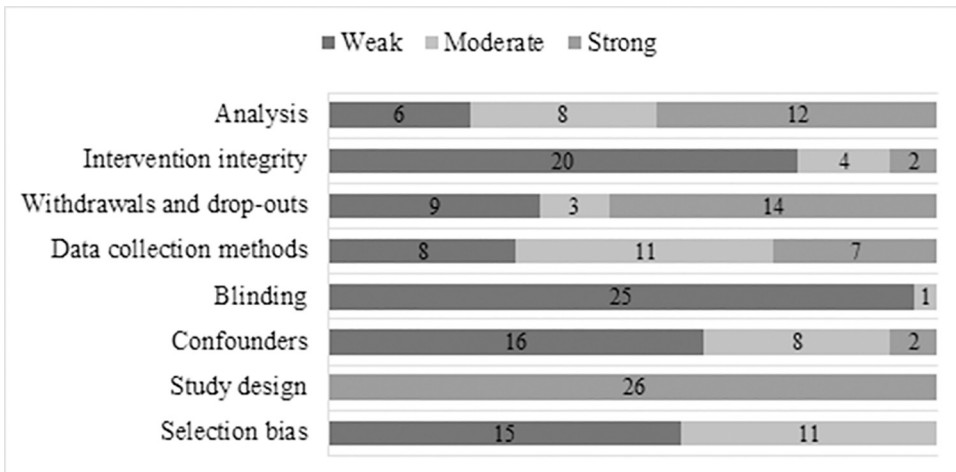

**Fig 2. Quality rating of items across included studies.**

presents the quality ratings of included studies as well as the effectiveness of interventions. S4 Table provides a complete overview of the outcomes of each study.

One study was rated as strong [56], three as moderate [39, 58, 59], and twenty-two as weak. Most weak scores were due to lack of 'blinding' (N = 25), not measuring 'intervention integrity' (N = 20) or lack of adjustment for 'confounders' (N = 17).

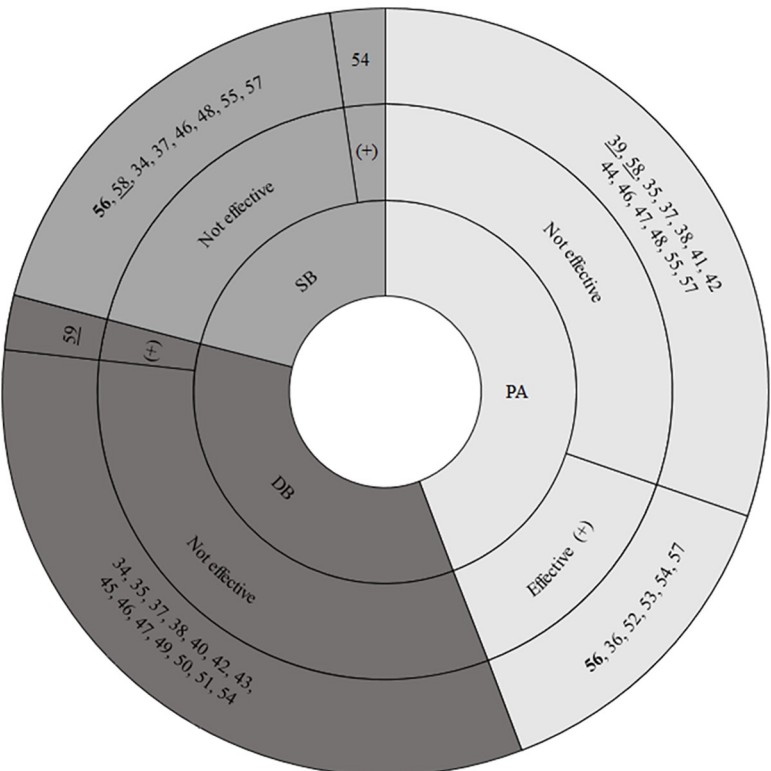

**Fig 3. Effectiveness of included studies in improving sedentary behavior, physical activity or dietary behavior.**
DB = dietary behavior, PA = physical activity, SB = sedentary behavior, (+) = study effective in improving the behavior. Bold number = high quality score, underscored number = moderate quality score, other numbers = low quality score. Of study 57 the fundamental movement skills-component was effective in improving PA, the behavioral modification-component was not.

## Effectiveness of the intervention

Six studies evaluated effects on physical activity [36, 39, 41, 44, 52, 53]; seven on dietary behavior [40, 43, 45, 49–51, 59]; five on physical activity and sedentary behavior [48, 55–58]; four studies on physical activity and dietary behavior [35, 38, 42, 47]; one study on dietary behavior and sedentary behavior [34]; and three studies on physical activity, sedentary behavior and dietary behavior [37, 46, 54]. No difference was found between the effectiveness of studies targeting one or multiple behaviors. Seven studies found improvements in at least 75% of the outcomes on physical activity, dietary behavior or sedentary behavior and we labelled those studies as 'effective'. Thirteen studies found improvements in less than 75% of the outcomes on one of the behaviors and six did not find any beneficial intervention effects. The study duration of the effective studies varied from four weeks [36] to two school years [52, 53], and the number of participants from 54 to 959 [59]. Notably, of the seven studies we defined as 'effective', two had a weak quality score on the validity and reliability of data collection methods [36, 54], four had a moderate score [53, 56, 57, 59] and only one a strong score [52]. Not one study had a strong quality score on selection bias, five scored moderate and two scored weak.

Six out of eighteen studies found beneficial effects on physical activity (see Fig 3) [36, 52–54, 56, 57]. One strong quality study evaluated an intervention aimed at improving physical activity at the expense of screen time by implementing ten lessons emphasizing self-monitoring, budgeting of time and selective viewing, and introducing children to street games. Children improved their number of thirty minute blocks/day in moderate-to-vigorous activity but no significant effects were found on sedentary behavior [56]. In one weak-quality study, a twelve-week school-based program aimed at increasing knowledge and understanding the benefits of a healthy diet and physical activity, improved children's time spent in light, moderate and vigorous physical activity [54]. Two weak-quality studies evaluated the same comprehensive school approach and improved children's daily physical activity level (steps/day) [52, 53]. One weak-quality study, aimed at promoting active transport to school, improved the percentage of daily commutes by cycling and moderate-to-vigorous physical activity [36]. One weak-quality study targeting children's fundamental movement skills [57], found significant effects on moderate-to-vigorous physical activity and counts per day, at post-intervention and at follow-up.

Out of nine studies evaluating effects on sedentary behavior, one weak-quality study evaluating a 12-lesson program on the importance of physical activity and healthy nutrition found beneficial effects on sedentary behavior [54]. Only one out of fifteen studies that evaluated effects on dietary behavior demonstrated significant beneficial effects. This moderate quality study evaluated a water campaign at schools, and demonstrated significant beneficial effects on parent-reported intake and servings of sugar-sweetened beverages and the observation report showed a reduction in percentage of sugar-sweetened beverages brought to school [59].

## Behavior change techniques

In all interventions, BCTs were identified and categorized according the BCT Taxonomy v1. In total, forty BCTs from this BCT Taxonomy were used in the included studies. We also identified BCTs that did not match any of the BCTs in the BCT Taxonomy, therefore three additional BCTs were added: 'Knowledge transfer' when new information was provided to children without a specific strategy or aim, 'Community involvement' when the community was involved in the development or delivery of the intervention, and 'Active learning' when several active teaching methods were included such as interactive games. Table 2 provides an overview of the grouped BCTs identified in the included studies. S4 Table provides a complete

**Table 2. Behavior change techniques (grouped) identified in the included studies.**

| Behavior Change Techniques[1] (93), Author, Year | Goals and planning | Feedback and Monitoring | Social Support | Shaping Knowledge | Natural Consequences | Comparison of Behavior | Associations | Repetition and Substitution | Comparison of Outcomes | Reward and Threat | Regulation | Antecedents | Identity | Scheduled Consequences | Self-Belief | Covert Learning | Knowledge Transfer | Active learning | Community involvement | Total (19) |
|---|---|---|---|---|---|---|---|---|---|---|---|---|---|---|---|---|---|---|---|---|
| Alaimo et al., 2015 [43] | | | | ✓ | | ✓ | ✓ | ✓ | | | | ✓ | | | | | ✓ | | | 6 |
| Ashfield-Watt et al., 2008 [51] | | | | | | | | | | | | ✓ | | | | | | | ✓ | 2 |
| **Bastian et al., 2015 [52]** | | | | ✓ | | ✓ | | ✓ | | | | ✓ | | | | | ✓ | | ✓ | 6 |
| Beyler et al., 2014 [44] | ✓ | | | ✓ | | ✓ | | ✓ | | | | | | | | | | | ✓ | 5 |
| Bohnert and Ward, 2013 [42] | | | | ✓ | | ✓ | | ✓ | | ✓ | | ✓ | | | ✓ | | ✓ | | ✓ | 8 |
| **Breslin et al., 2012 [54]** | | | | ✓ | ✓ | ✓ | | ✓ | ✓ | | | | | | | | | | | 5 |
| Colin-Ramirez et al., 2010 [55] | | | ✓ | ✓ | | ✓ | | ✓ | ✓ | | | ✓ | | | | | ✓ | | | 7 |
| Dunton et al., 2015 [41] | | | | | | | | | | ✓ | | | | | | | | | | 1 |
| Gatto et al., 2017 [49] | | | | ✓ | | ✓ | | ✓ | | | | ✓ | | | | | ✓ | | ✓ | 6 |
| Gittelsohn et al., 2010 [40] | | | ✓ | ✓ | | ✓ | ✓ | ✓ | | ✓ | | | | | | | ✓ | | ✓ | 8 |
| **Harrison et al., 2006 [56]** | ✓ | ✓ | ✓ | ✓ | | ✓ | ✓ | ✓ | | ✓ | | | | ✓ | | | ✓ | | ✓ | 11 |
| Keihner et al., 2017 [35] | ✓ | ✓ | ✓ | ✓ | ✓ | | ✓ | ✓ | | | | ✓ | | | | | ✓ | ✓ | | 10 |
| Lent et al., 2014 [45] | | | | | ✓ | | | | | | | ✓ | | | | | ✓ | | ✓ | 4 |
| Madsen et al., 2013 [39] | | | | ✓ | | ✓ | | ✓ | | | | | | | | | | | | 3 |
| **Mendoza et al., 2017 [36]** | | | ✓ | ✓ | | ✓ | | ✓ | | ✓ | | ✓ | | | | | ✓ | | | 7 |
| Neumark-Sztainer et al., 2009 [46] | | | ✓ | ✓ | ✓ | ✓ | ✓ | ✓ | | | | | ✓ | | | | ✓ | | ✓ | 9 |
| Nollen et al., 2014 [34] | | ✓ | | | | ✓ | ✓ | | | ✓ | | | | | | | | | | 4 |
| **Salmon et al., 2008 (FMS)[2] [57]** | | | | ✓ | | ✓ | | ✓ | | | | | | | | | | | | 3 |
| Salmon et al., 2008 (BM)[3] [57] | ✓ | | | ✓ | ✓ | ✓ | | ✓ | ✓ | ✓ | | ✓ | ✓ | ✓ | | | ✓ | ✓ | ✓ | 13 |
| Salmon et al., 2011 [58] | | | | | | | | | | | | | | | | | ✓ | | | 1 |
| Slusser et al., 2010 [38] | | | ✓ | ✓ | | ✓ | | ✓ | | | | ✓ | | | | | | | | 5 |
| Springer et al., 2012 [47] | ✓ | ✓ | ✓ | | | | | ✓ | | ✓ | | ✓ | | | | | | | ✓ | 7 |
| Trude et al., 2018 [50] | ✓ | | ✓ | ✓ | | ✓ | ✓ | ✓ | ✓ | ✓ | | ✓ | | | | | ✓ | ✓ | ✓ | 12 |
| **Van de Gaar et al., 2014 [59]** | | | ✓ | | ✓ | | ✓ | ✓ | ✓ | | | ✓ | | | | | ✓ | ✓ | ✓ | 9 |
| **Vander Ploeg et al., 2014 [53]** | | | ✓ | ✓ | | ✓ | | ✓ | | | | ✓ | | | | | ✓ | | ✓ | 7 |
| Wang et al., 2019 [37] | ✓ | ✓ | | ✓ | | ✓ | | ✓ | | ✓ | | ✓ | | | | | ✓ | ✓ | ✓ | 10 |
| Wells et al., 2014 [48] | | | | ✓ | | ✓ | | ✓ | | | | ✓ | | | | | ✓ | | | 5 |
| **Total (27)** | 9 | 6 | 9 | 19 | 6 | 20 | 7 | 22 | 5 | 10 | 0 | 16 | 2 | 2 | 1 | 0 | 18 | 5 | 16 | |

[1] Within each group several BCTs; the BCTs the outcomes on physical activity, dietary behavior and/or sedentary behavior. *Italic* are the newly added BCTs. **Bold** are studies that are effective in changing physical activity, dietary behavior and/or sedentary behavior. BM = behavioral modification, FMS = fundamental movement skills.

overview of all BCTs identified in the included studies. The most popular BCTs were *Behavioral practice/rehearsal* (N = 21), *Instruction on how to perform a behavior* (N = 19), and *Demonstration of the behavior* (N = 18). These three BCTs were often coded together, as in many intervention settings they complemented each other. The newly added BCTs, *Knowledge transfer* (N = 18) and *Community involvement* (N = 16) were also popular. *Adding objects to the environment* was applied in fourteen studies. For example, interventions provided healthy food for children, a garden for the school or extra information or objects were added to grocery stores to highlight healthy products. Interventions were not always described in detail, which limited the identification of the used BCTs.

A variety of BCTs were identified in the seven studies that showed beneficial effects on health behavior. No major differences were found between identified BCTs in effective versus non-effective interventions. The most popular BCTs used in effective interventions were *Behavioral practice/rehearsal*, *Instruction on how to perform a behavior*, *Demonstration of the behavior* (N = 6), and *Knowledge transfer* (N = 5), but these were also common in non-effective interventions.

## Discussion

This review provides an overview of interventions aiming to improve physical activity, sedentary behavior and dietary behavior in 9–12 year old children from low socioeconomic environments.

We considered an intervention effective when a beneficial intervention effect was obtained on at least 75% of the outcomes within a behavior (physical activity, dietary behavior or sedentary behavior). Using this definition, four out of six physical activity interventions were effective, one out of five physical activity and sedentary behavior interventions, one out of seven dietary behavior interventions, and one out of three studies focusing on physical activity, sedentary behavior and dietary behavior. Thirteen of the included studies found beneficial effects in less than 75% of the outcomes and six studies found no significant effects. All effective interventions focused on one or multiple health behaviors. All but one of the effective interventions took place during school hours. Reasons for the effectiveness of interventions during school hours could be that children spend a significant part of their day at school and children are more likely to take part in the intervention if it is included in the school curriculum [60]. Unfortunately, in most of the studies included in this review the participation rate was not adequately reported, nor was the number of participants that received the intended intervention or how consistently children participated throughout the intervention.

In our review only four studies had an overall strong or moderate quality rating. All but one study scored low on blinding of participants and assessors. Blinding may not be easy in health promotion studies, but even if this item would be omitted or studies would have included blinding, it would leave most of the studies with a weak quality rating. Six weak-quality studies would become moderate quality and one moderate-quality study would become high quality. More important is that many studies had small sample sizes and high attrition rates, limiting the power of studies and increasing the risk of attrition bias [36–38, 40–43, 45, 49, 50, 57]. Finding ways to improve recruitment of children from low socioeconomic environments and increasing parental involvement and consent, was often mentioned as a challenge and remains a point of attention for future studies [46, 47, 49, 57, 59]. Future studies should also include more valid and reliable data collections methods, as only seven studies included in this review scored high on this item. Studies also mentioned that longer-term interventions are warranted because the limited effectiveness of their intervention might be due to the short duration of the intervention (i.e. 4–16 weeks) [34–36, 49, 51, 56]. Thus, more high quality studies are needed to gain insights into promising BCTs for children from low socioeconomic environments.

No difference in BCTs used in effective and non-effective interventions was identified in our review, similar to a previous review focusing on obesity prevention and treatment interventions in adolescents from disadvantaged backgrounds [12]. Moreover, in the current review BCTs identified in the included studies targeting children from low socioeconomic environments were similar to BCTs identified in previous reviews that included studies focusing on the general population of children [11, 61, 62]. Another review that used a BCT Taxonomy to evaluate obesity prevention interventions in the general population of 2–18 year olds, concluded that *Generalizing behavior* was included in all four effective interventions [63], while this strategy was not present in the non-effective interventions. Therefore *Generalizing behavior*–which aims to encourage children to implement a behavior that was successful in one setting also in another setting [64]–seems a promising strategy to further examine in future interventions. *Giving general info*, *Rewards* and *Social comparison* were present in all four non-effective interventions suggesting that these BCTs are not sufficient for behavior change [63]. In our review *Knowledge transfer* and *Rewards* were part of effective as well as non-effective interventions, *Generalizing behavior* was part of three non-effective interventions and *Social comparison* was included in one effective intervention. More high quality research is needed to identify which BCTs are most effective for children in low socioeconomic environments, and how to effectively implement these BCTs, as implementation of is dependent on the local context [65]. As BCTs applied in effective interventions did not differ from those applied in non-effective interventions, the process of implementation may be key in successful behavior change. Further understanding is needed about how BCTs have their effects–i.e. their mechanisms of action–and how they should be implemented [66, 67]. This should also be studied specifically in interventions targeting children in low socioeconomic environments, to see whether different BCTs have to be applied or modes of implementation.

A possible explanation for the lack of evidence for effective BCTs, could be the low level of community participation in the development, implementation and evaluation of the intervention [68]. Even though many studies involved the community in the delivery of their intervention, few studies actively participated with the community and/or children in the development, implementation, and/or evaluation of the intervention. The lack of active involvement of the target group in this process may limit the support for and ownership of an intervention [69]. For example, corner stores did not have the capacity to store fresh items or fresh/healthier items were perceived as too expensive [45], intervention materials were not child-appropriate [40], or reimbursement of a physical activity program was not attractive because families did not have the finances to cover the costs up front [41]. One study described that the cooking and gardening intervention was culturally tailored by working with recipes that reflected foods prepared in the household of that community [49]. However, the questionnaire used in the effect evaluation of this study did not include fruits and vegetables that were commonly consumed in that specific community, leading to biased outcomes. Tailoring interventions to a specific community–by collaborating with the target group in intervention development and implementation–might lead to interventions that are more suitable to the needs and interests of the target group, creating more support for and ownership over the intervention, thereby potentially increasing its effectiveness. The target group can also be actively involved in the evaluation of the intervention, to make sure data collection methods are suitable and data is interpreted correctly [70].

Only three studies in our review involved children, parents or other community members to some extent in the development or implementation of the intervention [37, 40, 43]. Participatory studies rarely have a controlled design [71, 72], which may explain the lack of participatory studies in this review. Involving the target group already from the start of the intervention development may lead to better tailored and thereby more effective interventions. This may be

specifically the case for people from lower socioeconomic environments and minority groups that are generally not represented among intervention developers [73, 74]. By involving them, interventions could become more suitable to their needs and interests and help in a more thorough understanding of relevant barriers and facilitators of health behaviors in the study population [75, 76]. For example, food access and physical activity options in the community; the financial situation of the household; and norms, beliefs, culture and preferences about health behaviors [14, 76]. Moreover, being involved in the intervention development can increase the feeling of agency and leadership [77], which can have a positive influence on ownership, adherence and thus the effectiveness of the intervention [75]. Future research should compare the effectiveness of top-down developed interventions and interventions developed together with the target group and/or local stakeholders in a controlled design to explore the added value of co-creation in intervention development.

Another explanation for the lack of evidence for effective BCTs could be related to the delivery of interventions [39, 58]. Many interventions are in their delivery dependent on the commitment of schools or organizations whose primary task is not implementation of the intervention [78]. Commitment of the people and organizations that deliver the intervention is important for successful implementation and needs to be better evaluated and reported [79]. More knowledge of which BCTs are effective for which target group, could promote optimal use of BCTs.

This review has several strengths and limitations. A strength of this review is that the extraction of BCTs, data and the quality assessment was independently done by two researchers. The BCTs were structurally identified using the BCT Taxonomy v1, providing a thorough overview of the included studies and its content. We added three BCTs to this taxonomy, as we identified techniques that we could not link to any of the listed BCTs. Two of the added BCTs–Knowledge transfer and Community involvement–were frequently used. It must be noted that Community involvement may also encompass implementation strategies. Moreover, based on our review we cannot draw conclusions on the effectiveness of a single BCT but only on the effectiveness of the intervention as a whole. Another strength of this review is the focus on children from low socioeconomic environments which is important to gain more insights in effective BCTs for this high risk and hard to reach target group. Our review is restricted by the low number of studies (N = 26) making it difficult to draw conclusions on effective BCTs. Another limitation is that we score an intervention as effective based on the percentage of outcomes that were beneficially affected. As a result, studies that only report beneficial intervention effects are scored as more effective than studies that also report finding null findings. We therefore encourage authors to present both positive, negative and null findings. Moreover, the low number of studies (n = 4) with a moderate or strong quality rating hinders drawing strong conclusions. Lastly, a limitation is that a meta-analysis was not appropriate because of the heterogeneity in reported outcomes, intervention strategies and intervention duration between studies.

## Conclusions

Only seven out of 26 interventions in this review–of which one of high methodological quality–found significant beneficial effects on physical activity, sedentary behavior or dietary behavior. Secondly, both effective and non-effective interventions used similar BCTs. Moreover, BCTs applied in studies included in our review targeting children from low socioeconomic environments were similar to BCTs applied in studies targeting children from the general population included in previous reviews. A possible solution for more effective interventions that are better tailored to the specific circumstances, needs and interests of the target

group, may be co-creating interventions in collaboration with the children themselves as well as relevant stakeholders. This needs further research in both effectiveness studies comparing co-created interventions with top-down implemented interventions as well as implementation studies using appropriate evaluation designs.

## Supporting information

**S1 Checklist. PRISMA 2009 checklist.**
(PDF)

**S1 Table. Search strategy applied in Pubmed.**
(DOCX)

**S2 Table. Quality assessment tool.**
(DOCX)

**S3 Table. Quality assessment scores of included studies and its items.** Bold are studies that are effective in changing physical activity, dietary behavior and/or sedentary behavior. BM = behavioral modification, FMS = fundamental movement skills, + = strong score, +/- = moderate score - = weak score.
(DOCX)

**S4 Table. Behavior change techniques and outcomes of the included articles, sorted by methodological quality.** ↔ indicates intervention overall not effective on at least 75% of the outcomes within the behavior, ↑ indicates intervention overall effective on at least 75% of the outcomes within the behavior,? = unclear on the amount of outcome measures in the questionnaire. BM = behavioral modification, CG = control group, DB = dietary behavior, FMS = fundamental movement skills, FV = fruits and vegetables, IG = intervention group, MPA = moderate physical activity, MVPA = moderate to vigorous physical activity, PA = physical activity, SB = sedentary behavior, ST = screen time, T0 = baseline, VPA = vigorous physical activity, vs = versus. *ST is reported when a study specifically focused on ST instead of on SB in general.
(DOCX)

## Acknowledgments

The study was designed by MA, MC and TA. MA and TA reviewed the articles and extracted data, MA and MC conducted the quality assessment and MA and DA defined BCTs used in the included studies. The paper was drafted by MA, with all authors providing feedback to drafts. All authors approved the final version. The authors declare to have no competing interests.

## Author Contributions

**Conceptualization:** Manou Anselma, Mai J. M. Chinapaw, Teatske M. Altenburg.

**Formal analysis:** Manou Anselma, Mai J. M. Chinapaw, Daniëlle A. Kornet-van der Aa, Teatske M. Altenburg.

**Methodology:** Manou Anselma.

**Writing – original draft:** Manou Anselma.

**Writing – review & editing:** Mai J. M. Chinapaw, Daniëlle A. Kornet-van der Aa, Teatske M. Altenburg.

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
