## [Decision Letter · Decision Letter 0]

7 May 2020

PONE-D-20-01593

Effectiveness and promising strategies of interventions targeting energy balance-related behaviors in children from lower socioeconomic environments: a systematic review

PLOS ONE

Dear Ms. Anselma,

Thank you for submitting your manuscript to PLOS ONE. After careful consideration, we feel that it has merit but does not fully meet PLOS ONE’s publication criteria as it currently stands. Therefore, we invite you to submit a revised version of the manuscript that addresses the points raised during the review process.

We would appreciate receiving your revised manuscript by Jun 21 2020 11:59PM. To enhance the reproducibility of your results, we recommend that if applicable you deposit your laboratory protocols in protocols.io, where a protocol can be assigned its own identifier (DOI) such that it can be cited independently in the future. For instructions see: http://journals.plos.org/plosone/s/submission-guidelines#loc-laboratory-protocols

We look forward to receiving your revised manuscript.

Kind regards,

Rebecca E. Hasson, Ph.D

Academic Editor

PLOS ONE

2. Please report fully the results of the quality assessment performed in a Table, illustrating how each included study scored in every item of the tool used.

3. Our internal editors have looked over your manuscript and determined that it is within the scope of our Health Inequities and Disparities Research Call for Papers. This collection of papers is headed by a team of Guest Editors for PLOS ONE: Clare Bambra, Hans Bosma, Diana Burgess, Joseph Telfair, Barbara Turner, and Jennie Popay. The Collection will encompass a diverse range of research articles on health inequities and disparities.  Additional information can be found on our announcement page: hhttps://collections.plos.org/s/health-inequities

If you would like your manuscript to be considered for this collection, please let us know in your cover letter and we will ensure that your paper is treated as if you were responding to this call. If you would prefer to remove your manuscript from collection consideration, please specify this in the cover letter.

Reviewers' comments:

Reviewer's Responses to Questions

**Comments to the Author**

1. Is the manuscript technically sound, and do the data support the conclusions?

Reviewer #1: Yes

Reviewer #2: Partly

2. Has the statistical analysis been performed appropriately and rigorously? 

Reviewer #1: N/A

Reviewer #2: Yes

3. Have the authors made all data underlying the findings in their manuscript fully available?

Reviewer #1: No

Reviewer #2: Yes

4. Is the manuscript presented in an intelligible fashion and written in standard English?

Reviewer #1: Yes

Reviewer #2: Yes

5. Review Comments to the Author

Reviewer #1: The Systematic Review provides an interesting overview of intervention studies that adress children from low socioeconomic environments and aim to change their energy-balance related behaviors such as physical activity, dietary intake, and sedentary behavior. The authors summarized the (un-)effectiveness of the included studies and provide a thorough overview of the behavior change techniques that were included in the studies. In sum, it is an interesting and sound article adressing a very relevant topic by focussing children with low SES. However, their are some shortcomings and ambiguities.

Please finde my comments in detail:

1) I was wondering if the aim was to summarize the effectiveness for interventions implemented in low socioeconomic environments or for interventions addressing children with a low socioeconomic status. In my view these are two slightly different things. The search term looks like the authors searched for a target group / socioeconomic position and not for the “environment” where the intervention was implemented. Therefore, I was wondering if interventions that were implemented in a medium or mixed socioeconomic environment but targeted children with low SES were excluded? Was the aim to identify effective strategies for these environments or for children with low SES? I think it would help if the authors clarify this issue throughout the manuscript and if the authors provide an explanantion why they focus on low socioeconomic environments and not on children with a low SES position.

In my opinion, if the focus is on low socioeconomic environments, this should be described in the introduction more explicitly related to interventions that were implemented in these environments compared to interventions implemented in medium or mixed socioeconomic environments.

In the results the use of BCTs within the studies is described which is a very good idea, but I suppose to highlight this more in the abstract, introduction etc. Furthermore, the authors should use the term behavior change techniques consistently according to Michie and not mix it up with behavior change strategies or intervention strategies. This should be checked throughout the manuscript. Furthermore, it would be good, to highlight and explain why it is important to examine differences regarding effective behavior change techniques for this target group.

Methods

The authors provided a PRISMA checklist, but PRISMA is not mentioned in the Methods. It could be added that the review adheres to PRISMA

Results

Related to my comment above, what is meant in line 123 by “living in ‘low socioeconomic areas” or…” -> this gives the impression to me, that children living in those areas were addressed in this review but not the implementation in such an area. Even if it becomes clear in the table it would be good to describe and address this issue more consistently and carefully.

Related to this issue too, where only children included in the studies that have a low SES or took children with medium SES part as well? If implemented in low SE environments, the the latter would be the case depending from the indicator of low SE environment. So did the studies that were included in the review focused on the implementation in a low SE environment or addressing low SES children (which might be the same but which might also be not the same)? Could the authors provide the percentage of low SES children in the studies?

The table 2 is really interesting but much too long to be integrated in the main text. For the reader it is very difficult to read this table. I think it is not necessary to describe the BCTs for every study in detail, better would be an overview of the mostly used BCTs or something like this. This table would be good as an additional file.

It would be good to provide the reference number in the table, which makes it easier for the reader to combine text and table

Was there a difference in the effectiveness for studies addressing only one behavior compared to studies addressing two or three behaviors?

Please be consistent with writing out numbers or not, e.g. 4 week to 2 schools (line 148) but seven studies in line 149

Line 175 please add (BCTT) after the term Behavior Change Technique Taxononmy

What exactly means “Knowledge transfer” and doesn’t it fit into BCTT group 4 – Shaping Knowledge

In my opinion ‘Community involvement’ is not a BCT, it is more a kind of implementation strategy or a strategy that improves implementation (which is related to the effectiveness regarding behavior change).

I guess that ‘Active learning’ such as interactive games comprises BCTs such as rewards or social comparison etc.

Discussion

The sentence in line 198-200 is confusing and difficult to understand, it should be revised.

line 221-223: definite conclusions on effectiveness of intervention strategies is only possible when the strategies are implemented and evaluated separately. Studies – even high quality studies - such as those included in this review cannot draw conclusions on effectiveness of strategies (which would be important, too) only on effectiveness of the intervention as a whole. The authors should make this more clear. See also my comment above (consistent use of terms throughout the manuscript, e.g. 299)

Line 224 – 227: “we also found…” this is confusing as it reads such this was a part of the results

Line 238-239: "..., and how to effectively implement…" -> this is a new point which is very important and this issue should be introduced and discussed in detail

Related to the above mentioned issue of implementation: If the BCTs in effective and non-effective interventions did not differ, it might be that not the included BCTs are important but their implementation. Furthermore, if the BCTs for children with low SES and children from the general population are similar, what could be conclusions for addressing low SES people and develop interventions for this target group? There have to be differences, which are perhaps not the BCTs but strategies to reach this target group etc. It would be good, to discuss some ideas.

line 243: …participated… -> collaborated? what is meant by uptake and ownership? what about the relation to inappropriate measurements? It is difficult to understand what should be said with this sentence.

Line 303: This needs further research in high quality studies -> what is meant by this sentence? Are only high quality studies – implying RCTs etc. – necessary to address the above mentioned aspects? In my opinion more studies focussing the implementation are necessary, too. It would be good to add what these high quality studies should exactly adress and which other studies would help to further this area of research escpecially for this important target group.

Reviewer #2: The manuscript entitled 'Effectiveness and promising strategies of interventions targeting energy balance-related behaviors in children from lower socioeconomic environments: a systematic review' adresses an important topic. A comprehensive literature search was conducted and intervention strategies were extracted according to the Behavior Change Taxonomy v1. However, some comments should be adressed to improve the clarity of the manuscript.

Abstract:

- line 5: I would suggest to delete the term primary school and to only mention the age range. For example, in my country children aged six to ten years go to primary school.

- I would also suggest to include the information that you used the Behavior Change Taxonomy v1 to categorize intervention strategies.

Introduction:

- line 32: Do you mean "health inequalities between children from lower AND HIGHER socioeconomic positions"?

- lines 35-39: What about multi-level interventions? What does the evidence say about the effectiveness of interventions that adress both healthy eating and physical activity versus those that target only one behavior? Can you please give an example of an intervention study that improved physical education and was effective in terms of obesity prevention? Reference (16) focuses on policies but you are writing about obesity prevention interventions. Reference (11) only summarized the evidence reported in systematic reviews on the effectiveness of population-level childhood obesity prevention interventions that had an environmental component. Please revise that sentence.

- It is interesting to look at only interventions that targeted children from low socioeconomic background. But what was your hypothesis regarding your manuscript? Could you please better describe the rationale for your study? Did you except that you would find other intervention strategies that are associated with intervention effects in your target group compared to the whole population? Why did you expect that? Are there differences in terms of determinants of the three health behaviors that have been examined in other studies?

Methods:

- line 60: Again, I would suggest to delete the term "primary school". In line 45, you only write about the age range. Please be consistent.

- line 74: Please include one sentence about the full text screening.

Results:

- line 153: It is hard to follow your results section. There are some possibilities to present results from a systematic review in other formats than tables and text. E.g. you could think about doing a harvest plot. This would improve readability of your results section. Or maybe yoy could include more subheadings.

- line 186: Again, in tems of the results according to the Behavior Change Taxonomy, Table 2 does not give a good overview on this topic. For the reader it could be easier to have a bar chart that represents how often a technique was identified in the studies.

- line 190: "no major differences were found between identifies strategies in effective versus non-effective interventions". Please provide some numbers or a bar chart. Table 2 is not sufficient to give an overview on any differences between effective and non-effective interventions in terms of BCTs.

Discussion:

- line 200: "Thirteen studies found...". Thirteen studies of what?

- line 202: While it might be true that "effective" interventions focused on one or multiple health behaviors this might not be the case for obesity prevention interventions that focused on e.g. BMI as the outcome. Please include findings from other reviews and state whether your findings are in line with findings presented in other reviews or not.

- line 250: "Tailoring interventions to a specific community might increase effectiveness." How is that linked to your results?

6. PLOS authors have the option to publish the peer review history of their article (what does this mean?). If published, this will include your full peer review and any attached files.

Reviewer #1: Yes: Christina Niermann

Reviewer #2: Yes: Berit Steenbock

---

## [Author Response · Author response to Decision Letter 0]

3 Jul 2020

Before replying to the specific comments of the two reviewers, we would like to thank the reviewers for dedicating time to our work. From the reviewers’ remarks we conclude that some sections of our manuscript need additional clarification and/or revision. We feel that the reviewers’ constructive feedback enabled us to considerably improve our manuscript. Hereafter, we reply to the specific comments of the reviewers and have yellow marked the changes in our manuscript.

- Reviewer #1: The Systematic Review provides an interesting overview of intervention studies that address children from low socioeconomic environments and aim to change their energy-balance related behaviors such as physical activity, dietary intake, and sedentary behavior. The authors summarized the (un-)effectiveness of the included studies and provide a thorough overview of the behavior change techniques that were included in the studies. In sum, it is an interesting and sound article addressing a very relevant topic by focusing children with low SES. However, there are some shortcomings and ambiguities.

- Please find my comments in detail:

1) I was wondering if the aim was to summarize the effectiveness for interventions implemented in low socioeconomic environments or for interventions addressing children with a low socioeconomic status. In my view these are two slightly different things. The search term looks like the authors searched for a target group / socioeconomic position and not for the “environment” where the intervention was implemented. Therefore, I was wondering if interventions that were implemented in a medium or mixed socioeconomic environment but targeted children with low SES were excluded? Was the aim to identify effective strategies for these environments or for children with low SES? I think it would help if the authors clarify this issue throughout the manuscript and if the authors provide an explanation why they focus on low socioeconomic environments and not on children with a low SES position.

In my opinion, if the focus is on low socioeconomic environments, this should be described in the introduction more explicitly related to interventions that were implemented in these environments compared to interventions implemented in medium or mixed socioeconomic environments.

Our response: We thank the reviewer for addressing this important distinction. Our aim was to identify interventions that were effective in improving health behaviors in children from a low socioeconomic position (SEP). However, health promotion interventions targeting children from low SEP are usually implemented at a school- or community-level, where the school or community is identified as having an above-average percentage of children from low SEP. We elaborated on this in the introduction (lines 77-79): “An important note is that these interventions target children attending schools or living in neighborhoods defined as ‘disadvantaged’ or ‘low-income’, indicating that a relatively high percentage of children in these schools or neighborhoods have a low socioeconomic position.”

- In the results the use of BCTs within the studies is described which is a very good idea, but I suppose to highlight this more in the abstract, introduction etc. Furthermore, the authors should use the term behavior change techniques consistently according to Michie and not mix it up with behavior change strategies or intervention strategies. This should be checked throughout the manuscript. Furthermore, it would be good, to highlight and explain why it is important to examine differences regarding effective behavior change techniques for this target group.

Our response: The reviewer raised a very good point and we adapted the terminology as suggested. We also elaborated on the use of behavior change techniques in the abstract (lines 24-26) and introduction (lines 82-86):

Abstract:

“This systematic review aims to summarize the evidence regarding the effectiveness of interventions targeting energy balance-related behaviors in children from lower socioeconomic environments and the applied behavior change techniques.“

Introduction:

“A second aim was to identify effective behavior change techniques using the Behavior Change Technique (BCT) Taxonomy v1(28). Knowledge of BCTs used in interventions that are effective in improving health behaviors in children from low socioeconomic environments is important to inform future intervention development and improve the health of the children who mostly need it.”

- Methods

The authors provided a PRISMA checklist, but PRISMA is not mentioned in the Methods. It could be added that the review adheres to PRISMA.

Our response: We added a statement in the methods that the PRISMA guidelines were used (lines 89-90): “The Preferred Reporting Items for Systematic reviews and Meta-Analyses (PRISMA) statement was used to plan, conduct and transparently report this systematic review(29).”

- Results

Related to my comment above, what is meant in line 123 by “living in ‘low socioeconomic areas” or…” -> this gives the impression to me, that children living in those areas were addressed in this review but not the implementation in such an area. Even if it becomes clear in the table it would be good to describe and address this issue more consistently and carefully.

Related to this issue too, where only children included in the studies that have a low SES or took children with medium SES part as well? If implemented in low SE environments, the latter would be the case depending from the indicator of low SE environment. So did the studies that were included in the review focused on the implementation in a low SE environment or addressing low SES children (which might be the same but which might also be not the same)? Could the authors provide the percentage of low SES children in the studies?

Our response: We thank the reviewer for pointing out this ambiguity. Our aim was to identify interventions that were effective in improving health behaviors in children from a low socioeconomic position (SEP). However, health promotion interventions targeting children from low SEP are usually implemented at a school- or community-level, where the school or community is identified as having a relatively high percentage of children from low SEP. Studies rarely assess children’s individual SEP. The indicators that studies used for low SEP schools/communities are reported in Table 1, column 3. For example, a number of studies took place in schools where ≥50% of students qualified for free or reduced-price meal, implying that these studies also included children that did not qualify for these meals, potentially children with medium or high SEP. We added a clarification in the introduction (lines 77-79): “An important note is that these interventions target children attending schools or living in neighborhoods defined as ‘disadvantaged’ or ‘low-income’, indicating that a substantial percentage of children in these schools or neighborhoods have a low socioeconomic position.”

- The table 2 is really interesting but much too long to be integrated in the main text. For the reader it is very difficult to read this table. I think it is not necessary to describe the BCTs for every study in detail, better would be an overview of the mostly used BCTs or something like this. This table would be good as an additional file.

It would be good to provide the reference number in the table, which makes it easier for the reader to combine text and table

Our response: We thank the reviewer for the suggestions and moved the table to the Supporting Information (S4 Table) and added the reference numbers. We have added a new Table 2 to provide a more visual overview of the results regarding the BCTs. 

- Was there a difference in the effectiveness for studies addressing only one behavior compared to studies addressing two or three behaviors?

Our response: No difference was found between the effectiveness of studies targeting one or multiple behaviors. This is added in lines 194-195: “No difference was found between the effectiveness of studies targeting one or multiple behaviors.”

- Please be consistent with writing out numbers or not, e.g. 4 week to 2 schools (line 148) but seven studies in line 149

Our response: We apologize for the inconsistency and adjusted this. 

- Line 175 please add (BCTT) after the term Behavior Change Technique Taxononmy

Our response: We added (BCT) Taxonomy in line 83.

- What exactly means “Knowledge transfer” and doesn’t it fit into BCTT group 4 – Shaping Knowledge

Our response: We thank the reviewer for bringing up this point. BCT Taxonomy group 4 ‘Shaping Knowledge’ includes four behavior change techniques. However, the identified behavior change techniques related to shaping knowledge did not match any of these four BCTs: as the identified techniques did not 1) instruct how to perform a behavior, 2) provide information about antecedents, 3) re-attribute behavior and causes, or 4) include behavioral experiments. We therefore added the broader category ‘knowledge transfer’ similar to the review of Kornet-van der Aa et al. (2017), to be able to categorize techniques that described transferring information from person A to person B without providing further details on the mode or goal. We specified this in the results section (lines 230-236): “In all interventions, BCTs were identified and categorized according the BCT Taxonomy v1. In total, forty BCTs from this BCT Taxonomy were used in the included studies. We also identified BCTs that did not match any of the BCTs in the BCT Taxonomy, therefore, three additional BCTs were added: ‘Knowledge transfer’ when new information was provided to children without a specific strategy or aim, ‘Community involvement’ when the community was involved in the development or delivery of the intervention, and ‘Active learning’ when several active teaching methods were included such as interactive games.”

- In my opinion ‘Community involvement’ is not a BCT, it is more a kind of implementation strategy or a strategy that improves implementation (which is related to the effectiveness regarding behavior change).

Our response: We understand that the reviewer suggests ‘community involvement’ to be more of an implementation strategy. However, involvement of the community can also take place during intervention development. We added in the discussion that the added behavior change technique can also encompass an implementation strategy (lines 354-355): “It must be noted that Community involvement may also encompass implementation strategies.”

- I guess that ‘Active learning’ such as interactive games comprises BCTs such as rewards or social comparison etc.

Our response: We agree with the reviewer that this could be the case, however, the included studies did not specify for which aim they implemented such methods. Studies for example mentioned that interactive games were implemented, but did not explicitly clarify the aim. Therefore, we decided to add a separate category active learning.

- Discussion

The sentence in line 198-200 is confusing and difficult to understand, it should be revised.

Our response: We apologize that this sentence was not clear. We have added an example in the results section (lines 124-126): “For example, if a study measured eight different outcomes related to physical activity, six had to show a beneficial intervention effect for the study to be considered effective in improving physical activity.”

We have also rewritten the mentioned sentence in the discussion (lines 254-262): “We considered an intervention effective when a beneficial intervention effect was obtained on at least 75% of the outcomes within a behavior (physical activity, dietary behavior, sedentary behavior). Using this definition, four out of six physical activity interventions were effective, one out of five physical activity and sedentary behavior interventions, one out of seven dietary behavior interventions, and one out of three studies focusing on physical activity, sedentary behavior and dietary behavior.”

- line 221-223: definite conclusions on effectiveness of intervention strategies is only possible when the strategies are implemented and evaluated separately. Studies – even high quality studies - such as those included in this review cannot draw conclusions on effectiveness of strategies (which would be important, too) only on effectiveness of the intervention as a whole. The authors should make this more clear. See also my comment above (consistent use of terms throughout the manuscript, e.g. 299)

Our response: We thank the reviewer for this valuable note and we have added the following in the discussion (lines 282-284 and 355-356): “Thus, more high quality studies are needed to gain insights into promising BCTs for children from low socioeconomic environments.” 

“Moreover, based on our review we cannot draw conclusions on the effectiveness of a single BCT but only on the effectiveness of the intervention as a whole.”

- Line 224 – 227: “we also found…” this is confusing as it reads such this was a part of the results

Our response: We have rewritten the sentence (lines 287-290): “Moreover, in the current review BCTs identified in the included studies targeting children from low socioeconomic environments were similar to BCTs identified in previous reviews that included studies focusing on the general population of children(11, 62, 63).”

- Line 238-239: "..., and how to effectively implement…" -> this is a new point which is very important and this issue should be introduced and discussed in detail

Our response: We thank the reviewer for this suggestion and have added more detail (lines 299-305): “More high quality research is needed to identify which BCTs are most effective for children in low socioeconomic environments, and how to effectively implement these BCTs, as implementation of BCTs is dependent on the local context(66). As BCTs applied in effective interventions did not differ from those applied in non-effective interventions, the process of implementation may be key in successful behavior change. Further understanding is needed about how BCTs have their effects – i.e. their mechanisms of action – and how they should be implemented(67, 68).”

- Related to the above mentioned issue of implementation: If the BCTs in effective and non-effective interventions did not differ, it might be that not the included BCTs are important but their implementation. Furthermore, if the BCTs for children with low SES and children from the general population are similar, what could be conclusions for addressing low SES people and develop interventions for this target group? There have to be differences, which are perhaps not the BCTs but strategies to reach this target group etc. It would be good, to discuss some ideas.

Our response: We appreciate the reviewer for thinking along and have added our ideas on these matters in lines 

302-307: “As BCTs applied in effective interventions did not differ from those applied in non-effective interventions, the process of implementation may be key in successful behavior change. Further understanding is needed about how BCTs have their effects – i.e. their mechanisms of action – and how they should be implemented(67, 68). This should also be studied specifically in interventions targeting children from low socioeconomic environments, to see whether different BCTs have to be applied or different modes of implementation.”

- line 243: …participated… -> collaborated? what is meant by uptake and ownership? what about the relation to inappropriate measurements? It is difficult to understand what should be said with this sentence.

Our response: We apologize that this sentence was not clear. We have rewritten it as follows (lines 310-313): “Even though many studies involved the community in the delivery of their intervention, few studies actively participated with the community and/or children in the development, implementation, and/or evaluation of the intervention. The lack of active involvement of the target group in this process may limit the support for and ownership of an intervention(70).”

- Line 303: This needs further research in high quality studies -> what is meant by this sentence? Are only high quality studies – implying RCTs etc. – necessary to address the above mentioned aspects? In my opinion more studies focusing the implementation are necessary, too. It would be good to add what these high quality studies should exactly address and which other studies would help to further this area of research especially for this important target group.

Our response: We fully agree with the reviewer that not per se only RCTs are necessary and that implementation studies are needed as well. We elaborated on this in lines(376-378): “This needs further research in both effectiveness studies comparing co-created interventions with top-down implemented interventions as well as implementation studies using appropriate evaluation designs.”

- Reviewer #2: The manuscript entitled 'Effectiveness and promising strategies of interventions targeting energy balance-related behaviors in children from lower socioeconomic environments: a systematic review' addresses an important topic. A comprehensive literature search was conducted and intervention strategies were extracted according to the Behavior Change Taxonomy v1. However, some comments should be addressed to improve the clarity of the manuscript.

Abstract:

- line 5: I would suggest to delete the term primary school and to only mention the age range. For example, in my country children aged six to ten years go to primary school.

Our response: We thank the reviewer for this suggestion and understand that ‘primary school’ is not applicable to all countries. We therefore removed this term.

- I would also suggest to include the information that you used the Behavior Change Taxonomy v1 to categorize intervention strategies.

Our response: We thank the reviewer for this valuable suggestion and have added it to the abstract (lines 24-26 and 31-33): “This systematic review aims to summarize the evidence regarding the effectiveness of interventions targeting energy balance-related behaviors in children from lower socioeconomic environments and applied behavior change techniques. “

“Two independent researchers extracted data, identified behavior change techniques using the Behavior Change Technique Taxonomy v1 […]”

Introduction:

- line 32: Do you mean "health inequalities between children from lower AND HIGHER socioeconomic positions"?

Our response: We indeed meant ‘lower and higher’ and have changed it accordingly. 

- lines 35-39: What about multi-level interventions? What does the evidence say about the effectiveness of interventions that address both healthy eating and physical activity versus those that target only one behavior? Can you please give an example of an intervention study that improved physical education and was effective in terms of obesity prevention? Reference (16) focuses on policies but you are writing about obesity prevention interventions. Reference (11) only summarized the evidence reported in systematic reviews on the effectiveness of population-level childhood obesity prevention interventions that had an environmental component. Please revise that sentence.

Our response: Indeed interventions focusing on multiple behaviors and system levels have shown to be more effective. We have added this to this paragraph and included new references to support this argument. We have also included the mentioning of ‘policies’ to specify the type of interventions. The section is now written as follows (lines: 60-69): “Previous systematic reviews focused on children from all socioeconomic positions(11), on adolescents(12), on children from a specific ethnicity(13, 14) or were limited to specific intervention designs such as family-based(15), school-based(16), or policy interventions(17). Effective components of obesity prevention interventions in children identified in systematic reviews include school policies regarding the availability of foods and beverages meeting nutritional standards; targeting multiple behaviors and system levels; encouragement of environments and cultural practices at school and home that support healthy behavior; education of children, parents and teachers on healthy nutrition and physical activity; improvement of physical education programs and physical activity possibilities in policy and practice(11, 16-20).”

- It is interesting to look at only interventions that targeted children from low socioeconomic background. But what was your hypothesis regarding your manuscript? Could you please better describe the rationale for your study? Did you except that you would find other intervention strategies that are associated with intervention effects in your target group compared to the whole population? Why did you expect that? Are there differences in terms of determinants of the three health behaviors that have been examined in other studies?

Our response: We hypothesized that different behavior change techniques would be used in effective versus non-effective studies. We now more clearly described this aim in the introduction (lines 82-86): “A second aim was to identify effective behavior change techniques using the Behavior Change Technique (BCT) Taxonomy v1(28). Knowledge on BCTs used in interventions that are effective in improving health behaviors in children from low socioeconomic environments is important to inform future intervention development and improve the health of the children who mostly need it.”

As we did not find differences in applied behavior change techniques, the mode of implementation may be more important, and we added this to in the discussion (lines 299-307): “More high quality research is needed to identify which BCTs are most effective for children in low socioeconomic environments, and how to effectively implement these BCTs, as implementation of BCTs is dependent on the local context(66). As BCTs applied in effective interventions did not differ from those applied in non-effective interventions, the process of implementation may be key in successful behavior change. Further understanding about which BCTs are preferred by children, how BCTs have their effects – i.e. their mechanisms of action – and how they should be implemented(67, 68). This should also be studied specifically in interventions targeting children in low socioeconomic environments, to see whether different BCTs have to be applied or that their low effectiveness for example comes from a misunderstanding of the mechanisms of action or modes of implementation.”

Concerning the latter question the reviewer poses, differences have indeed been identified in determinants of certain behaviors. We have added the following in the introduction (lines 69-71): “Previous studies have also shown that energy-balance related behaviors and its determinants may manifest themselves differently in children from different socioeconomic levels(21-25).”

Methods:

- line 60: Again, I would suggest to delete the term "primary school". In line 45, you only write about the age range. Please be consistent.

Our response: We apologize for the inconsistency and we have deleted the term ‘primary school’ throughout the paper.

- line 74: Please include one sentence about the full text screening.

Our response: We included the following sentences (lines 111-113): “Full texts were screened by MA, and TA or DA. In case of discrepancies or uncertainties, a third and/or fourth reviewer was consulted.”

Results:

- line 153: It is hard to follow your results section. There are some possibilities to present results from a systematic review in other formats than tables and text. E.g. you could think about doing a harvest plot. This would improve readability of your results section. Or maybe you could include more subheadings.

Our response: We agree with the reviewer that the results section could benefit from visual alternatives to text and tables. We therefore removed Table 2 and added it as Supporting Information (S4 Table). We have added Figs 2 and 3 and a new Table 2 to provide a more visual overview of the results. We have also rewritten line 153 into (lines 204-207): “Six out of eighteen studies found beneficial effects on physical activity (36, 52-54, 56, 57). One strong quality study evaluated an intervention aimed at improving physical activity at the expense of screen time by implementing ten lessons emphasizing self-monitoring, budgeting of time and selective viewing, and introducing children to street games.”

- line 186: Again, in terms of the results according to the Behavior Change Taxonomy, Table 2 does not give a good overview on this topic. For the reader it could be easier to have a bar chart that represents how often a technique was identified in the studies.

- line 190: "no major differences were found between identifies strategies in effective versus non-effective interventions". Please provide some numbers or a bar chart. Table 2 is not sufficient to give an overview on any differences between effective and non-effective interventions in terms of BCTs 

Our response: We have created a new table showing the (grouped) behavior change techniques – according to the BCT Taxonomy v1 – identified in the included studies (Table 2), and in bold highlighted the effective interventions.

Discussion:

- line 200: "Thirteen studies found...". Thirteen studies of what?

Our response: We apologize that this sentence was not clear. We have reworded it as follows (lines 262-263): “Thirteen of the included studies found beneficial effects in less than 75% of the outcome measures and six studies found no significant effects.”

- line 202: While it might be true that "effective" interventions focused on one or multiple health behaviors this might not be the case for obesity prevention interventions that focused on e.g. BMI as the outcome. Please include findings from other reviews and state whether your findings are in line with findings presented in other reviews or not.

Our response: We thank the reviewer for this suggestion. For the current review, we chose to focus specifically on interventions that aimed at improving behaviors instead of BMI and therefore also felt it was not appropriate to compare our findings to interventions that focused on BMI as an outcome. For the comparisons with other reviews focusing on improving energy-balance related behaviors we kindly refer to lines 285-299.

- line 250: "Tailoring interventions to a specific community might increase effectiveness." How is that linked to your results?

Our response: We have rewritten this section to provide more context and linking it to our results (lines 312-326): “The lack of active involvement of the target group in this process may limit the support for and ownership of an intervention(70). For example, corner stores did not have the capacity to store fresh items or fresh/healthier items were perceived as too expensive(45), intervention materials were not child-appropriate(40), or reimbursement of a physical activity program was not attractive because families did not have the finances to cover the costs up front(41). One study described that the cooking and gardening intervention was culturally tailored by working with recipes that reflected foods prepared in the household of that community(49). However, the questionnaire used in the effect evaluation of this study did not include fruits and vegetables that were commonly consumed in that specific community, leading to biased outcomes. Tailoring interventions to a specific community – by collaborating with the target group in intervention development and implementation – might lead to interventions that are more suitable to the needs and interests of the target group, creating more support for and ownership over the intervention, thereby potentially increasing its effectiveness. The target group can also be actively involved in the evaluation of the intervention, to make sure data collection methods are suitable and data is interpreted correctly(71).”

---

## [Decision Letter · Decision Letter 1]

7 Aug 2020

Effectiveness and promising behavior change techniques of interventions targeting energy balance-related behaviors in children from lower socioeconomic environments: A systematic review

PONE-D-20-01593R1

Dear Dr. Anselma,

We’re pleased to inform you that your manuscript has been judged scientifically suitable for publication and will be formally accepted for publication once it meets all outstanding technical requirements.

Kind regards,

Rebecca E. Hasson, Ph.D

Academic Editor

PLOS ONE

Reviewers' comments:

Reviewer's Responses to Questions

**Comments to the Author**

1. If the authors have adequately addressed your comments raised in a previous round of review and you feel that this manuscript is now acceptable for publication, you may indicate that here to bypass the “Comments to the Author” section, enter your conflict of interest statement in the “Confidential to Editor” section, and submit your "Accept" recommendation.

Reviewer #1: All comments have been addressed

Reviewer #2: All comments have been addressed

2. Is the manuscript technically sound, and do the data support the conclusions?

Reviewer #1: Yes

Reviewer #2: Yes

3. Has the statistical analysis been performed appropriately and rigorously? 

Reviewer #1: N/A

Reviewer #2: N/A

4. Have the authors made all data underlying the findings in their manuscript fully available?

Reviewer #1: Yes

Reviewer #2: Yes

5. Is the manuscript presented in an intelligible fashion and written in standard English?

Reviewer #1: Yes

Reviewer #2: Yes

6. Review Comments to the Author

Reviewer #1: The authors did a great job in revising their manuscript. The authors addressed all my comments, provided good answers and made adequate changes in the manuscript. I do not have further comments.

Reviewer #2: (No Response)

7. PLOS authors have the option to publish the peer review history of their article (what does this mean?). If published, this will include your full peer review and any attached files.

Reviewer #1: **Yes: **Christina Niermann

Reviewer #2: **Yes: **Berit Brandes